

**Knowledge needs, available actions and future challenges in agricultural soils**

Georgina Key[1*], Mike G. Whitfield[2], Julia Cooper[3], Franciska T. De Vries[1], Martin Collison[4], Thanasis Dedousis[5], Richard Heathcote[6], Brendan Roth[7], S.A.M. Shamal[8], Andrew Molyneux[9], Wim H. Van der Putten[10], Lynn. V. Dicks[11], William. J. Sutherland[11], Richard D. Bardgett[1].

[1]Faculty of Life Sciences, Michael Smith Building, The University of Manchester, Oxford Road, Manchester, M13 9PL, UK

[2]Lancaster Environment Centre, Lancaster University, Lancaster, LA1 4YQ

[3]School of Agriculture, Food and Rural Development, Newcastle University, Kings Road, Newcastle Upon Tyne, NE1 7RU

[4]Collison and Associates Limited, Honeysuckle Cottage,  Shepherdsgate Road, Tilney All Saints, King's Lynn, Norfolk, PE34 4RW

[5]European Agro Development Team, PepsiCo Europe

[6]Richard Heathcote, R&J Sustainability Consulting Ltd, 21 Lattimore Road, Stratford upon Avon, CV37 0RZ, working with: National Association of Cider Makers, Cool Farm Alliance, and Innovate UK

[7]Department for Environment, Food & Rural Affairs, Nobel House, 17 Smith Square, London, SW1P 3JR

[8]GeoInfo Fusion Ltd, Cranfield, Bedford, MK43 0DG

[9]Huntapac Produce Ltd, 293 Blackgate Lane, Holmes, Tarleton, Preston, Lancashire, PR4 6JJ

[10]Netherlands Institute of Ecology, Department of Terrestrial Ecology and Laboratory of Nematology, Wageningen University and Research Centre, Droevendaalsesteeg 10, 6708 PB Wageningen

[11]Department of Zoology, University of Cambridge, Cambridge CB2 3QZ, UK.

*Correspondence to: G. Key (georgina.key@ahdb.org.uk)

**Abstract**

The goal of this study is to clarify research needs and identify effective actions for enhancing soil health. This was done by a synopsis of soil literature that specifically tests actions designed to maintain or enhance elements of soil health. Using an expert panel of soil scientists and practitioners, we then assessed the evidence in the soil synopsis to highlight actions beneficial to soil health, actions considered detrimental, and actions that need further investigation. Only seven of the 27 reviewed actions were considered to be beneficial, or likely to be beneficial in enhancing soil health. These included the use of a mixture of organic and





inorganic soil amendments, cover crops, crop rotations, and growth of crops between crop rows or underneath the main crop, and the use of formulated chemical compounds (such as nitrification inhibitors), the control of traffic and traffic timing, and reducing grazing intensity. Using a partial Spearman's correlation to analyse the panel's responses, we found that

increased certainty in scientific evidence led to actions being considered to be more effective due to them being empirically justified. This suggests that for actions to be considered effective and put into practice, a substantial body of research is needed to support the effectiveness of the action. This is further supported by the high proportion of actions (33%), such as changing the timing of ploughing or amending the soil with crops grown as green manures, that experts

felt had unknown effectiveness, usually due to insufficient robust evidence. Our assessment, which uses the Delphi technique, increasingly used to improve decision-making in conservation and agricultural policy, identified actions that can be put into practice to benefit soil health. Moreover, it has enabled us to identify actions that need further research, and a need for increased communication between researchers, policy-makers and practitioners, in

order to find effective means of enhancing soil health.

**Key words:** Soil health; Delphi technique; Systematic review;

## 1. Introduction

Soil health in agro-ecosystems describes the continued ability of a soil to sustain crop (or animal) growth over the long-term through efficient recycling and provision of nutrients and water, and is controlled by a variety of factors, including soil physical and chemical properties, soil organic matter, and the activities of diverse soil biological communities (Maeder et al 2002; Bardgett 2005; Barrios, 2007; Lamarque et al. 2011). Enhancing soil health is central to

delivering food security and ecosystem services (Lal 2009; de Vries et al. 2012; Lipper et al, 2014). As agriculture has become increasingly intensified, and agro-ecosystems less biologically diverse, the ecosystem processes underpinning soil health are being eroded (Glover et al, 2010; Amundson et al, 2015; FAO and ITPS, 2015; Bardgett 2016). In addition to food production, healthy soils also underpin a wide range of ecosystem services, including

the carbon sequestration, flood control and biological control of pests and diseases (Lavelle et al 2005; Wall et al. 2012; FAO and ITPS, 2015; Bardgett 2016), which are crucial to underpinning sustainable development goals.

Soil degradation is caused by many factors, including deforestation, infrastructure development and construction, but inappropriate management of agricultural land is also a

major cause (Terranova et al 2009; Nunes et al 2011). For example, increased mechanisation and size of farm machinery has caused extensive soil compaction (a major factor in soil degradation) (Beylich et al 2010; Allman et al 2015); continuous tillage, which disrupts soil





structure and increases soil organic matter loss, has accelerated rates of soil erosion in parts of the world (Martinez-Casasnovas and Ramos, 2009; Don et al, 2010; Crittenden et al, 2015); heavy grazing by livestock leaves land sparsely vegetated, compacted, and vulnerable to soil erosion (Lal and Stewart 1990; Nunes et al 2011); and leaving cropland without a protective

vegetative cover causes declines in soil organic matter content, and leaves soil exposed to the erosive forces of wind and rain (Lal and Stewart 1990; Pimentel et al 1995). Evidence is accumulating that intensive farming practices reduce the diversity and complexity of soil food webs (Tsiafouli et al. 2015), which has consequences for the functioning of soil and its ability to buffer against extreme weather events (De Vries et al. 2012).

While many practitioners are well versed in how to maintain soil health, they are often not aware of the trade-offs that exist between enhancing certain soil properties and maintaining the functions that underpin them. For example, relatively little is known about how farming practices influence the diversity and functioning of complex soil microbial communities that are responsible for transforming nutrients into plant available forms, or what can be done

to harness the benefits of soil organism activities for soil health and crop production  (Philippot et al. 2013; Bardgett and van der Putten 2014). Also, management practices that have been shown to maintain many ecosystem services in tandem with soil fertility, such as mulching, composting and specific crop rotations, might not markedly benefit soil biodiversity (Turbé et al 2010). There is also much discussion about how to best manage for soil health, resulting in

the need for evidence-based environmental policies for sustainable soil management, as well as the identification of knowledge needs for researchers and practitioners.

The overall goal of this paper was to identify effective actions for enhancing soil health and clarify future research needs. This was done by synopsis of soil literature that specifically tests actions designed to maintain or enhance elements of soil health. Using an expert panel

of soil scientists and practitioners, we then assessed the evidence in the soil synopsis to highlight actions beneficial and detrimental to soil health, and actions that need further investigation. We used the Delphi technique (Mukherjee et al. 2015) to produce a ranked list of current evidence-based actions for enhancing soil health (Sutherland et al 2004, 2011). The Delphi technique is a data synthesis method that seeks to find a consensus between experts

on a particular subject (Hsu and Sandford 2007). It is widely used in medicine to clarify particular issues, assess gaps in knowledge, enhance decision-making, and inform policy (Jones and Hunter 1995; Hasson et al, 2000; Hsu and Sandford 2007). However, it also presents opportunities to improve decision-making in conservation and agricultural policy (Sutherland 2006). For example, the Delphi technique was used to determine a package of

'best available techniques' to reduce nitrogen emissions from poultry units (Angus et al 2003) and to quantify the effectiveness and certainty of evidence to determine beneficial actions for



conservation (Walsh et al. 2013). We used the Delphi technique to identify and assess actions that benefit soil health.

### 2. Methods

5       We identified major threats to soil health, such as erosion, compaction, nutrient leaching and biodiversity loss, using lists compiled by the UK Soil Association (Marmo 2012) and Scottish Environment Protection Agency (SEPA). To identify the scientific literature relevant to enhancing or maintaining soil health, we used two approaches: a literature search, using key search terms within a database, and a journal trawl whereby we examined every

published article and manually selected relevant papers. For the literature search we used the Web of Science database (Thomson Reuters 2014) and search terms were chosen using an iterative process of searching and refining. The terms used in this search focused on actions to maintain or restore natural (or semi-natural) soil processes related to soil health. The initial searches returned 37,748 hits. The first 100 titles for each search term were examined and

the search refined. Duplicate studies were removed. All article titles and abstracts were examined and irrelevant references excluded. A panel of experts were selected to help refine the number of studies. They were chosen based on their expertise in their respective fields, to give a range of perspectives on the literature, and to highlight potential issues either with the science or implementation of an action. Study abstracts for the remaining 543 references were

then scanned to identify studies meeting two criteria: (1) there was an action that farmers or land-managers could perform to enhance soil health on their land; and (2) effects of the action were monitored quantitatively. These criteria excluded studies examining the effects of specific actions without testing them explicitly. For example, predictive modelling studies and correlative studies were excluded (Dicks et al. 2013b).

For the journal trawl, seven journals were selected based on the wide scope of their soil-related research and on the recommendation of experts in soil science. These included: *European Journal of Soil Science, Geoderma, Global Change Biology, Land Use Policy, Soil Biology and Biochemistry, Soil Use and Management,* and *Journal of Applied Ecology.* Study titles and abstracts were scanned from volume one of each journal to mid-2012. The trawl

identified 175 studies relevant to all soil health actions. These literature review methods together returned a total of 718 studies monitoring the effects of actions in the list. As this was part of a project looking at how to increase food security, we included European studies and regions with similar temperate climates where similar agricultural practices were used. The majority of our papers relate to farming in temperate regions of the world, especially Europe

and the US; our references are therefore only a sample of the global literature, but nonetheless represent a substantial body of evidence and include a broad spectrum of journals that publish soil-related research.



The literature was distilled into a synopsis of actions for enhancing soil health, available online (www.nercsustainablefood.com, www.conservationevidence.com.) and in Key et al. (2015). The list of 27 actions to enhance soil health was developed from a list suggested by several academics who work in relevant fields, and who were not part of the Expert Panel (see

below). These actions were refined and added to as we reviewed the literature. Actions were included if they could realistically be adopted by farmers an land-managers, regardless of whether they had already been adopted, or whether or not evidence for their effectiveness already existed. All captured studies relevant to soil health were included.

The Expert Panel consisted of three experts from academia, three from private-sector

research, one from a governmental body, one from an agricultural consultancy and one from agri-business. All have expertise in soil research and land management, and each of them provided an independent assessment of the evidence for each action. They were asked to participate because they were either primary stakeholders or decision-makers, and because they all have specialised knowledge in soil management (Hsu and Sandford, 2007). Expert

panels range widely in number. In an assessment of the effectiveness of various conservation measures, expert panel numbers ranged from 4 to 47 members (Sutherland et al 2015), whereas in a systematic review of healthcare quality indicators the average number of panel members was 17 (Boulkedid et al, 2011). Here, we had nine panel members who were asked to allocate a score to each action using the online survey software Qualtrics

(www.qualtrics.com). Their assessment was based on four factors: the effectiveness of each action in enhancing soil health; certainty in the evidence for each action; the strength of potential negative side-effects associated with implementing the action; and finally soil types and locations covered. The panel were asked to ignore prior knowledge of effectiveness and base their scoring only on the evidence presented in the synopsis.

The Delphi technique was used to quantify effectiveness of the actions and certainty of evidence (Rowe and Wright 1999; Hutchings and Raine 2006). In this technique, the panel completes a repeated, anonymous survey of evidence to elicit an expert judgement on a complex problem (Mukherjee et al, 2014). The number of panel members used in the Delphi technique can vary, but the average of several experts' opinions is likely to be more reliable

than an individual assessment of a problem (Sutherland et al, 2015). This paper extends work by Sutherland et al (2011) in which the technique was applied to conservation in agro-ecosystems to promote evidence-based practice (Sutherland et al 2004). The Expert Panel members independently scored the four factors listed above using a percentage scale for each action for the first round of scoring, and then received the collated evidence from the rest of

the Expert Panel, with the aim of collaboratively refining the judgements of each panel member (Walsh et al 2013). The ability of panel members to see each other's (anonymous) comments can lead to a refining of opinions and allow the panel to approach decision-making using



another perspective (Hasson et al, 2000). Based on the differing perspectives encountered, each expert then entered final assessments for each of the practices and comments were recorded. One advantage of using Qualtrics is that final scores were not unduly influenced by dominant personalities (Sutherland 2006; Burgman et al. 2011). The order in which actions

were presented was varied to prevent panel bias in scoring from order of presentation. All scoring took place remotely via the Qualtrics website.

The 27 actions for enhancing soil health were then ranked according to the median of the final scores, as assessed by the expert panel. The scores were used to put the actions into six categories, following the method described by Sutherland et al (2015): (1) Beneficial (in

enhancing soil health); (2) Likely to be beneficial; (3) Trade-offs between benefits and adverse side-effects; (4) Unknown effectiveness; (5) Unlikely to be beneficial; and (6) Likely to be ineffective or to have adverse side-effects. The categories are based on threshold values of certainty, effectiveness and negative side-effects, i.e. on a combination of the benefit and harm and the strength of the evidence (Appendix A, Table 1).

Due to the relatively low number of actions, we used a partial Spearman's correlation to analyse the Expert Panel's assessment, to identify any relationship between the certainty of the evidence and the perceived effectiveness of each action. We used the ppcor package (Kim 2012) in R, version 3.1.1 (R Core Team, 2014). The median scores for the effectiveness of each action and certainty of evidence were the main variables, with the strength of potential

negative side effects from implementing the actions as the controlled variable.

### 3. Results

The 27 actions assessed by the Expert Panel were ranked by how beneficial each action is to soil health (Table 1).


**Table 1. The 27 actions for enhancing soil health ranked according to the median of final scores (1 – 100) by scientists, practitioners and policy-makers, from most beneficial through to harmful. The scores have been used to put the practices into six indicative categories, based on categories (Sutherland et al. 2015): 1. Beneficial; 2.**

**Likely to be beneficial; 3. Trade-offs between benefits and adverse side-effects; 4. Unknown effectiveness; 5. Unlikely to be beneficial; 6. Likely to be ineffective or to have adverse side-effects.**

| Action | Effectiveness | Certainty | Negative side-effects | Category |
|---|---|---|---|---|



| | | | | | |
|---|---|---|---|---|---|
| 1 | Amend the soil using a mix of organic and inorganic amendments | 69 | 64 | 15 | Beneficial |
| 2 | Grow cover crops when the field is empty | 75 | 67 | 16 | Beneficial |
| 3 | Use crop rotation | 66 | 75 | 8 | Beneficial |
| 1 | Grow cover crops beneath the main crop (living mulches) or between crop rows | 65 | 54 | 19 | Likely to be beneficial |
| 2 | Amend the soil with formulated chemical compounds | 64 | 46 | 19 | Likely to be beneficial |
| 3 | Control traffic and traffic timing | 55 | 62 | 18 | Likely to be beneficial |
| 4 | Reduce grazing intensity | 51 | 58 | 14 | Likely to be beneficial |
| 1 | Change tillage practices | 61 | 72 | 46 | Trade-offs |
| 2 | Convert to organic farming | 55 | 52 | 64 | Trade-offs |
| 3 | Amend the soil with manures and agricultural composts | 70 | 59 | 26 | Trade-offs |
| 4 | Add mulch to crops | 60 | 64 | 23 | Trade-offs |
| 5 | Retain crop residues | 63 | 54 | 29 | Trade-offs |
| 6 | Restore or create low input grasslands | 53 | 59 | 32 | Trade-offs |
| 7 | Amend the soil with municipal wastes or their composts | 45 | 44 | 54 | Trade-offs |
| 8 | Amend the soil with fresh plant material or crop remains | 53 | 53 | 34 | Trade-offs |
| 9 | Incorporate leys into crop rotation | 46 | 45 | 36 | Trade-offs |
| 10 | Plant new hedges | 49 | 45 | 20 | Trade-offs |





| | | | | |
|---|---|---|---|---|
| 1 | Change the timing of ploughing | 46 | 38 | 33 | Unknown effectiveness |
| 2 | Amend the soil with organic processing wastes or their composts | 58 | 35 | 20 | Unknown effectiveness |
| 3 | Change the timing of manure application | 50 | 33 | 24 | Unknown effectiveness |
| 4 | Amend the soil with crops grown as green manures | 53 | 36 | 16 | Unknown effectiveness |
| 5 | Amend the soil with composts not otherwise specified | 54 | 29 | 19 | Unknown effectiveness |
| 6 | Amend the soil with non-chemical minerals and mineral wastes | 35 | 37 | 23 | Unknown effectiveness |
| 7 | Amend the soil with bacteria or fungi | 40 | 31 | 17 | Unknown effectiveness |
| 8 | Use alley cropping | 36 | 23 | 19 | Unknown effectiveness |
| 9 | Encourage foraging waterfowl | 14 | 34 | 20 | Unknown effectiveness |
| 1 | Reduce fertilizer, pesticide use | 26 | 40 | 48 | Likely to be ineffective or harmful |

Of the 27 actions, only three were considered to be unequivocally beneficial to soil health, namely the use of a mix of organic and inorganic soil amendments, growing cover crops, and

5    crop rotation. The three actions found to be beneficial had the highest certainty and effectiveness scores, along with good coverage in the literature (Figure 1) and weak negative side effects. Four of the actions were considered likely to be beneficial, namely grow cover crops beneath the main crop (living mulches) or between crop rows; amend the soil with formulated chemical compounds; control traffic and traffic timing; and reduce grazing intensity.

10    However they received lower certainty an effectiveness scores, due in part to the smaller body of evidence available. The only action that fell into the 'likely to be ineffective or harmful' category was reducing fertiliser and pesticide use, largely due to consequent reductions in crop yields





Nine of the actions were scored as having 'unknown effectiveness.' This was largely because few of the studies captured for those actions were replicated or randomised, and therefore many panel members felt unable to comment on the effects of the actions with certainty. Other factors that contributed to 'unknown effectiveness' include fewer geographical
5    locations or soil types coverage.

For some actions, several of the practitioners on the Expert Panel were surprised that no negative effects had been reported and questioned why actions such as 'change the timing of ploughing' were not common practice if there were no negative side effects. For other actions, panel members identified known negative side effects from their own knowledge, and also
10    suggested additional studies, but these were not included in the assessment. The majority of these actions fell into the 'trade-offs' category, where evidence suggested that the actions were either beneficial in specific circumstances, or considered likely to be beneficial but with strong negative side effects.

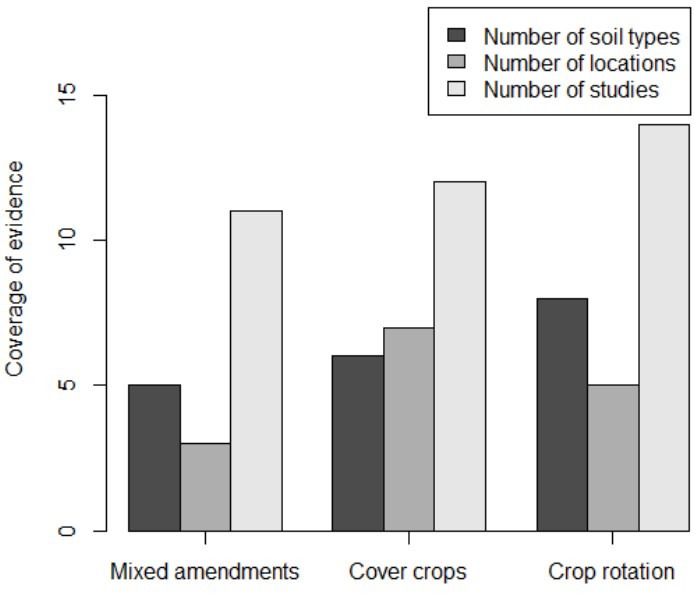

15         Figure 1. Coverage of soil type, geographical location and number of studies for the three actions identified as beneficial to soil fertility. Mixed amendments = amend with a mix of organic and inorganic amendments; cover crops = grow cover crops when field is empty; crop rotation = use crop rotation.



We accounted for the negative side effects of actions using a partial Spearman's Rank correlation analysis. We detected a significant positive relationship between the effectiveness of the action and certainty of the evidence ($r_{partial} = 0.72$, n= 27, $P$= <0.001) (Figure 2). There was no significant relationship between effectiveness and negative side-effects ($r_{partial} = -0.34$,

5      n= 27, $P$= 0.09), or certainty and negative side-effects ($r_{partial} = 0.17$, n= 27, $P$= 0.39).

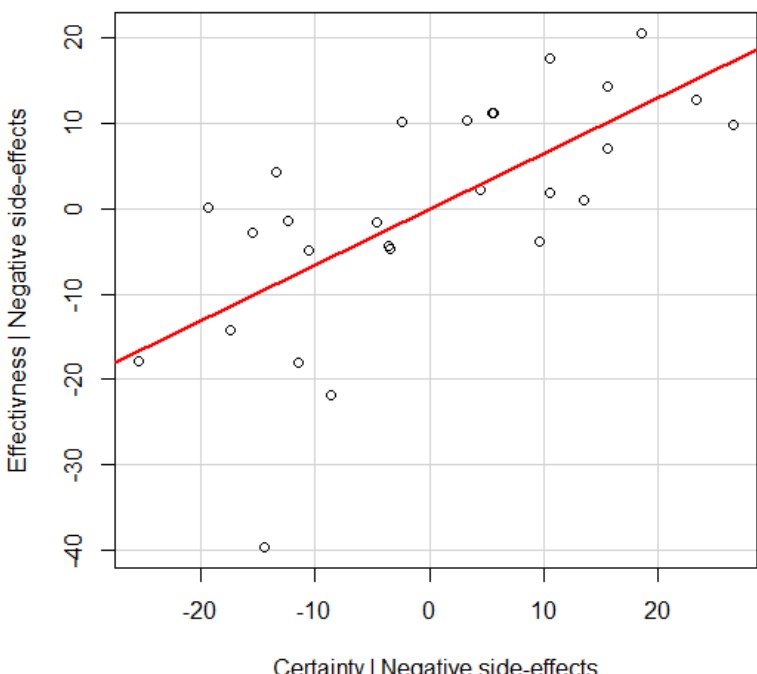

Figure 2. Partial Spearman's Rank correlation, determined using the median scores for each of the 27 actions given by the expert panel. Points are plotted according to the first two variables, effectiveness and certainty, while controlling for the third, negative side effects. Effectiveness = effectiveness of the action, certainty = certainty of the evidence, negative side-effects = negative side-effects of implementing the action. Spearman rank correlation coefficient $r_s = 0.72$, $P$ <0.001. The red line represents the slope of the least squares line for the residual series.

### 4. Discussion

To clarify research needs and identify effective actions for enhancing soil health, a

10     synopsis of soil literature was carried out and then assessed by experts in the field. The input from both scientists and practitioners helped to identify existing knowledge that should be made more accessible to those who put research into practice (Dicks et al, 2013a), and highlighted a wide spectrum of certainty regarding the actions covered in this review. The three





beneficial actions identified by this process, namely amend the soil using a mix of organic and inorganic amendments, grow cover crops when the field is empty, and use crop rotation, are well established and have been used for centuries to build soil health. The four actions likely to be beneficial to soil health in agro-ecosystems, namely grow cover crops beneath the main

crop or between crop rows, amend the soil with formulated chemical compounds, control traffic and traffic timing, and reduce grazing intensity, were considered to be effective, but warranted more evidence before the Expert Panel would state them as conclusively beneficial. The only action considered harmful to soil health was reducing fertilizer and pesticide use, largely due to its negative effect on crop yields, which the panel felt outweighed any positive effects of

reducing fertiliser and pesticide use on biodiversity (Tonitto et al, 2006; Foley et al, 2011Tscharntke et al, 2012).  Perhaps of most significance, however, are the 19 actions falling into the 'trade-offs' or 'unknown effectiveness' categories, which not only highlights the high level of uncertainty about most actions, but also the large number of current soil management practises that are based on non-scientific knowledge.

The beneficial actions, such as 'use crop rotation,' have considerable supporting evidence, and both scientists and practitioners are aware of the merits of these actions across several localities and soil types. The panel felt that there was clear evidence showing that crop rotation is beneficial to soil health, especially when legumes are included in the rotation (Blair & Crocker, 2000; Gregorich et al, 2001; Mäeder et al 2002; Schjønning et al, 2007; Ryan et

al, 2008). Including legumes in rotation can lead to other benefits, such as improved water filtration and reduced competition with weeds (Place et al 2003). The panel also considered that the evidence for this action covered a wide range of geographic locations and soil types, but showed interest in seeing more studies demonstrating the effect of type and length of rotation on soil biodiversity. Panel members felt that the action 'amend the soil using a mix of

organic or inorganic amendments' contributed to a wider range of nutrients and mineralisation processes occurring in the soil, which was also found by Palm et al (1997), as well as providing other benefits for crop productivity, such as weed reduction. This action also scored highly due to the range of soil types covered by the evidence. The panel considered the evidence for 'growing cover crops when the field is empty' to provide good coverage in Europe and scored

highly for the effectiveness of this action. Research has shown that cover cropping over winter can reduce soil and nutrient loss (Ding et al, 2006; Gülser, 2006; Zhang et al. 2007; Zhou et al, 2012). The research was supported by the panel who considered the action to be most appropriate to the UK when used over winter as post-harvest cover cropping. These Expert Panel responses, and the rich historical literature on the benefits of these actions for soil

health, are evidence that they can be regarded as 'hot topics' in this field, which are identified by Sutherland (2013) as areas of research that are progressing and, in the case of crop





rotation, widely implemented for many years (although less used now, or simplified in intensive systems) (Benton et al. 2003).

The 'likely to be beneficial' actions were generally considered to be effective with few negative side-effects, but the Expert Panel suggested that more evidence was needed,
particularly in the form of commercial evidence or case studies, or with specific effects on yields presented. The Spearman's rank correlation suggests that the relationship between negative side effects and the perceived effectiveness of the action is not as closely correlated as the relationship between effectiveness and certainty of evidence, where more of the variance in the data is accounted for (Chatterjee and Hadi 2012). The significant positive
relationship found between the effectiveness of the action and certainty of the evidence suggests that, for both scientists and practitioners, increased certainty in the scientific evidence presented for actions resulted in them being considered more effective.

A key finding of our assessment is that it is not yet clear how effective the majority of the reviewed actions are for enhancing soil health. Others have argued that the provision of
ecosystem services is limited by a lack of scientific understanding (e.g., Benayas et al 2009); likewise, our findings suggest that we do not yet have a full understanding of the consequences of actions on soil health. Of the 27 actions reviewed, we found the nine have 'unknown effectiveness', and ten have trade-offs to their implementation. For actions falling in the 'trade-offs' category, there are clear benefits to implementing them; however, they may
need to be refined to minimise any negative effects. For example, conversion to organic farming can increase soil organic matter and soil biodiversity (Liu et al, 2007; Birkhofer et al, 2008; Canali et al, 2009; Chaudhry et al, 2012), but can make protecting crops from pests and diseases more difficult, and result in lower yields. Planting new hedges reduces soil loss (Anigma et al, 2002; Mutegi et al, 2008; Donjadee & Tingsanchali, 2013), but could make
cultivation more difficult. Amending soil with manures and agricultural composts increases soil organic matter levels  (Jones et al, 2006; Celik et al, 2010; Bhattacharyya et al, 2012), but may need to be avoided close to water, due to possible increased nitrate leaching and subsequent water quality problems (Díez et al, 2004). Refining actions was suggested by the practitioners on the panel. For example, refining the action 'reduce fertilizer and pesticide use'
to question *which* pesticides and fertilizers should be reduced, and what rate, rather than having a blanket reduction, could reduce the trade-offs of such an action. Accessing knowledge from practitioners, which would not necessarily make it into scientific literature, such as the practical barriers to and the specific details of implementing actions, would add more context to the results in this paper. This highlights the importance of two-way knowledge
exchange if we are to effectively enhance soil health.

We are aware that the review method we used has limitations, for example only one literature database was used in addition to journal trawling. The Expert Panel suggested





studies that were not captured by our searches over the period of the project, and were surprised at there being so few papers for some of the actions. The journals trawled represent a spectrum of journals that publish soil-related research and are all well respected in the field. We recognise, however, that there is scope to extend our analysis to consider an even wider

range of literature including applied research by industry. The panel also highlighted actions not included in this synopsis that warrant further research, or suggested alternative actions, such as 'mob' grazing, where a field is heavily grazed, before removing the animals for a rest period (Bittman and MacCartney 1994). Our database of references is only a sample of the global literature, but with 132 papers reviewed it nonetheless represents a substantial amount

of evidence, and demonstrates what can be achieved within a short timeframe. Previous research has queried the use of the Delphi technique as a stand-alone decision-making tool (Angus et al, 2003), so the next step would be to expand the review process and Delphi methods (i.e. a larger Expert Panel with additional rounds of scoring) to capture the full breadth of available evidence for soil health (Rowe and Wright 2011). For some actions the range of

technology used, for example in 'formulated chemical amendments' highlights the difficulty in comparing studies; what might be suitable in one location on one soil type may not be appropriate for others. The condition of a site also needs to be taken into account when recommending practices, given that the impact of various actions will vary depending on many factors, including soil type, the extent that a soil is degraded, and local climate. Although not

useful for forming broad applications, reviews such as this could lead to targeted 'best-fit' approaches more beneficial to local soil health, an approach found by Giller et al (2010) to be better for different types of farms. The review also provides an informative starting point on appropriate actions for improved soil health.

## 5. Conclusions

This review provides a useful case study of a method to incorporate expert knowledge into the implementation of evidence-based practice to improve soil health in agro-ecosystems. Not only have we highlighted several ways of maintaining or improving soil health that are based on scientific evidence, but also we identify a high level of uncertainty surrounding many

interventions that are widely used to maintain soil health. Further, our assessment has also identified major research gaps and areas of uncertainty in relation to the effectiveness of certain interventions, which may prove to be a barrier to implementing actions. Expanding the scope of the review in future work may help to identify some of the uncertainty surrounding actions, and refine what further research is needed. Agricultural intensification is required to

improve food security, however this needs to be done in a sustainable way if we are to have more resilient agricultural systems in the face of climate change (Garnett et al, 2013). By implementing the beneficial actions as assessed above and addressing some of the gaps in



our knowledge, we could go some way to restoring functional biodiversity and associated ecosystem services for good soil health in agro-ecosystems.

**Author contribution**

The project was initiated by W.J. Sutherland and R.D Bardgett. G. Key and M. G. Whitfield carried out the systematic review, with input from R. D. Bardgett, L. V. Dicks and W. J. Sutherland. G. Key carried out the Delphi process with contributions from L. V. Dicks. Ranking was carried out by J. Cooper, F. De Vries, M. Collison, T. Dedousis, R. Heathcote, B. Roth, S.A.M. Shamal, A. Molyneux and W. Van der Putten, and collated by G. Key. G. Key carried out the statistical correlations with help from M. G. Whitfield. G. Key prepared the manuscript with contributions from all co-authors.

**Acknowledgments**

This work was supported by the Natural Environment Research Council [grant number NE/K001191/1]. WJS is funded by Arcadia. Thanks to Joscelyn Ashpole for her contributions to the process and to Dr Phil Donkersley for his statistical advice.

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
