# Peer review of "Knowledge needs, available practices and future challenges in agricultural soils"

_SOIL, 2016_

## Short Comment (SC1) · 15 May 2016

**General Comment**

This article is a useful contribution to the current evidence relating to a range of soil management practices in agricultural systems. My one concern is whether 'enhancing soil health' is the most appropriate paradigm in which to frame the study. In their

recent paper on the dimensions of soil security (McBratney *et al.*, 2014) stated that, 'there is little value in talking about the health of any given soil, unless there is an understanding of how 'healthy' it can actually be. Unlike capability, the condition [one of their five dimensions of soil security] of a soil is contemporary and is measured on a short-term management time scale'. [I would like to note here that I am not an author, nor did I contribute to, the paper on the dimensions of soil security]. The definitions of 'soil quality' and 'soil health' lack the wider framework of the dimensions of soil security and I would argue that your study largely addresses the 'soil condition' dimension, and it would have been better to frame it in this context which provides a more comprehensive framework for the discussion of the state of soil systems.

Reference: Alex McBratney, Damien J. Field, Andrea Koch, The dimensions of soil security, Geoderma, Volume 213, January 2014, Pages 203-213, ISSN 0016-7061 http://dx.doi.org/10.1016/j.geoderma.2013.08.013.

---

## Referee Comment (RC1) · I. Stavi (Referee) · 31 May 2016

Overall, the authors have identified an interesting topic, and have appropriately highlighted some of the current agricultural challenges, which deserve further investigation. The topic is important and relevant for increasing understanding regarding the goals of conservation agriculture and the efficiency of its methods.

Specifically, the authors mentioned the beneficial role of 'the use of a mixture of organic and inorganic soil amendments'. Indeed, integrated nutrient management encompasses an important part of integrated farming systems, which have been proven to sustain soil health. Yet, among the topics discussed in this paper, it seems that the topic of integrated farming systems deserves more investigation. On the one hand, over the last few years there is an increasing awareness of integrated and moderate-

intensity agricultural systems, and of their beneficial role in sustaining soil functions and ecosystem services. On the other hand, there is still a lot more to be discovered regarding these effects. Of particular interest in this regard is livestock grazing of crop stubble in mixed agro-pastoral systems, whose intensity was identified by the authors as a factor that has to be reduced. Yet, recent studies show that moderate stocking rates have improved soil quality and functioning over the long term. At the same time, it is clear that the topics included (or excluded) in this paper were determined by the very specific methodology utilized by the authors (as detailed in the methods section). It is therefore suggested that the authors should highlight this topic among the ones which deserve more research in the future.

Regardless, some modifications are recommended in order to make the paper clearer. For example, the term 'action' for describing agricultural practices is not so common. I'd recommend modifying this term to 'practice' throughout the paper. Similarly, I'd suggest the authors use acceptable and professional terms such as integrated nutrient management (instead of 'a mixture of organic and inorganic soil amendments'); cover cropping (instead of 'grow cover crops when the field is empty'); intercropping (instead of 'growth of crops between crop rows'); manuring and composting (instead of 'amend the soil with manures and agricultural composts'); mulching (instead of 'add mulch to crops'); and best management practices (BMPs) or recommended management practices (RMPs) (instead of 'best available techniques'). The authors may find several documentations of these professional terms, either in the Web of Science database or open sources.

In the abstract, it is not so clear what the seven beneficial practices are. I'd suggest numbering them to increase clarity (i.e., 'These included the use of (1) a mixture of organic and inorganic soil amendments [which should be replaced with 'integrated nutrient management']; (2) cover crops; (3) crop rotations; (4) the growth of crops between crop rows [which should be replaced with 'intercropping'] or underneath the main crop; (5) the use of formulated chemical compounds (such as nitrification inhibitors); (6) the

control of traffic and traffic timing; and (7) reducing grazing intensity').

In the beginning of the methods section, the authors are asked to provide some more information about the major threats to soil health they have identified. In the current version, the list seems to be very partial.

The results section is sufficiently informative.

In the discussion section, some modifications are needed regarding the use of inaccurate terms, such as 'cover crops between crop rows' (should be intercropping) and 'soil loss' (should be soil erosion). Apart from that, the bimodal (positive and negative) effects which define the 'trade-off category' further highlight the important role of integrated agricultural systems, which combine both conventional and conservation practices in order to decrease environmental footprint while sustaining crop yields productivity. Regardless, similar to integrated nutrient management, the practicing of integrated agricultural systems also includes the concept of integrated pest management (IPM). Among other missing topics in this study, the neglecting of this concept highlights the paper's limitations. I'd recommend the authors to specify this topic in the study's major limitations (the last paragraph of the discussion section).

Usually, no citations are allowed in the conclusions sections. The authors are asked to go through the journal instructions and check on this point.

---

## Author Comment (AC1) · 19 Jun 2016

Dear Barry,

Thank you for your comment, which raises an important point about framing the work in the context of 'soil health'. We felt that 'soil health' rather than 'soil condition' was appropriate as practitioners tend to view the latter as being just the physical condition of the soil, without taking into account the biological elements and the overall properties which emerge from the physical and biological elements interacting (Romig et al, 1996; Kibblewhite et al, 2008). Hence, we chose to use soil health because it portrays the soil as a dynamic, living system (Doran and Zeiss 2000), and we wanted to ensure that the living, biological element of soil was not excluded when the Expert Panel assessed the practices/actions. We accept that the phrase 'soil health' is often used loosely,

but it is being used in recent literature regarding best practices for soil management, including some that we cover in our paper (Han et al, 2016), and the term is used by the USDA Natural Resource Conservation Service Soils, and by the UK Government in their recently published Soil Health report (HC 180). Given this we feel that this term will resonate with a wider audience of practitioners and policy makers, which is a target audience of our paper.

Doran, J. W. & Zeiss, M. R.: Soil health and sustainability: managing the biotic component of soil quality. Applied Soil Ecology, 15, pages 3-11 (2000).

Han, P., Zhang, W., Wang, G., Sun, W. & Huang, Y.: Changes in soil organic carbon in croplands subjected to fertilizer management: a global meta-analysis. Nature (published online June 2016). http//:doi:10.1038/srep27199.

Kibblewhite, M. G., Ritz, K. & Swift, M. J.: Soil health in agricultural systems. Phil. Trans. R. Soc. B, 363, pages 685-701 (2008)

Romig, D. E., Garlynd, M. J. & Harris, R. F.: Farmer-based assessment of soil quality: a soil health scorecard Soil Science Society of America, 677 S (1996).

---

## Author Comment (AC2) · 19 Jun 2016

Dear Ilan,

Thank you for your comments, which highlight the importance of using recognised terminology and areas that this manuscript did not cover. We agree with your suggestions for changes in terminology, and have made your suggested changes throughout, which we believe will make the manuscript clearer. In particular we have modified the word 'action' for 'practice' to describe agricultural practices, the seven beneficial actions have been numbered in the abstract, and we have changed the terminology for the individual practices as recommended.

With respect to your comment that we should provide more information on major threats to soil health, we now include more detail, identifying the complete list of threats

to soil health identified by UK Soil Association (Marmo 2012), the Department for Environment, Food and Rural Affairs (Defra, 2009), and Scottish Environment Protection Agency (SEPA). In addition to the threats previously mentioned in the manuscript, secondary threats, including carbon loss, pollution and flooding, have been included in lines 13-15 on page 4. These were selected as being the most relevant threats to soil health in the UK and other temperate zones, which we now make clearer in our methods section (page 4).

Regarding your comment on the lack of integrated pest management covered in this manuscript, we did not include this because it was not identified as a major threat to soil health in our initial scoping exercise. However, as requested we have added text to our discussion (page 10) to identify the omission of this and other practices, such as stubble grazing, as a limitation of the study and as areas that warrant further research. We also stress that this study was part of a larger project reviewing practices that can deliver conservation benefits in agro-ecosystems, and aspects of IPM were considered by another group (see Sutherland et al 2015 for more detail).

As requested we checked our conclusions section and removed the one citation that we included (Garnett et al. 2013).

Please see the adjusted manuscript for changes.

William J. Sutherland, Lynn V. Dicks, Nancy Ockendon and Rebecca K. Smith, What Works in Conservation Series, vol. 1 | ISSN: 2059-4232 (Print); 2059-4240 (Online), 2015.

Please also note the supplement to this comment:
http://www.soil-discuss.net/soil-2016-17/soil-2016-17-AC2-supplement.pdf
* * *
[Figure]

**Supplement:**

**Knowledge needs, available practiceactions and future challenges in agricultural soils**

Georgina Key[1*], Mike G. Whitfield[2], Julia Cooper[3], Franciska T. De Vries[1], Martin Collison[4], Thanasis Dedousis[5], Richard Heathcote[6], Brendan Roth[7], S.A.M. Shamal[8], Andrew Molyneux[9], Wim H. Van der Putten[10], Lynn. V. Dicks[11], William. J. Sutherland[11], Richard D. Bardgett[1].

[1]Faculty of Life Sciences, Michael Smith Building, The University of Manchester, Oxford Road, Manchester, M13 9PL, UK

[2]Lancaster Environment Centre, Lancaster University, Lancaster, LA1 4YQ

[3]School of Agriculture, Food and Rural Development, Newcastle University, Kings Road, Newcastle Upon Tyne, NE1 7RU

[4]Collison and Associates Limited, Honeysuckle Cottage,  Shepherdsgate Road, Tilney All Saints, King's Lynn, Norfolk, PE34 4RW

[5]European Agro Development Team, PepsiCo Europe

[6]Richard Heathcote, R&J Sustainability Consulting Ltd, 21 Lattimore Road, Stratford upon Avon, CV37 0RZ, working with: National Association of Cider Makers, Cool Farm Alliance, and Innovate UK

[7]Department for Environment, Food & Rural Affairs, Nobel House, 17 Smith Square, London, SW1P 3JR

[8]GeoInfo Fusion Ltd, Cranfield, Bedford, MK43 0DG

[9]Huntapac Produce Ltd, 293 Blackgate Lane, Holmes, Tarleton, Preston, Lancashire, PR4 6JJ

[10]Netherlands Institute of Ecology, Department of Terrestrial Ecology and Laboratory of Nematology, Wageningen University and Research Centre, Droevendaalsesteeg 10, 6708 PB Wageningen

[11]Department of Zoology, University of Cambridge, Cambridge CB2 3QZ, UK.

*Correspondence to: G. Key (georgina.key@ahdb.org.uk)

**Abstract**

The goal of this study is to clarify research needs and identify effective actions practices for enhancing soil health. This was done by a synopsis of soil literature that specifically tests actions practices designed to maintain or enhance elements of soil health. Using an expert panel of soil scientists and practitioners, we then assessed the evidence in the soil synopsis to highlight actions practices beneficial to soil health, actions practices considered detrimental, and actions practices that need further investigation. Only seven of the 27 reviewed actions practices were considered to be beneficial, or likely to be beneficial in

enhancing soil health. These included the use of: (1) integrated nutrient management (organic and inorganic amendments); (2) cover crops; (3) crop rotations; (4)  intercropping between crop rows or underneath the main crop (5) formulated chemical compounds (such as nitrification inhibitors); (6)  control of traffic and traffic timing; and (7) reducing grazing intensity. Using a partial Spearman's correlation to analyse the panel's responses, we found that increased certainty in scientific evidence led to  practices being considered to be more effective due to them being empirically justified. This suggests that for  practices to be considered effective and put into practice, a substantial body of research is needed to support the effectiveness of the action. This is further supported by the high proportion of  practices (33%), such as changing the timing of ploughing or amending the soil with crops grown as green manures, that experts felt had unknown effectiveness, usually due to insufficient robust evidence. Our assessment, which uses the Delphi technique, increasingly used to improve decision-making in conservation and agricultural policy, identified  practices that can be put into practice to benefit soil health. Moreover, it has enabled us to identify  practices that need further research, and a need for increased communication between researchers, policy-makers and practitioners, in order to find effective means of enhancing soil health.

**Key words:** Soil health; Delphi technique; Systematic review; Evidence-based  practices

**Commented [GK1]:** Hereafter referred to as integrated nutrient management

**Commented [GK2]:** We feel that it is important to illustrate that both methods of intercropping were reviewed, so I have left this distinction in. Will refer to it as 'intercropping' throughout the rest of the manuscript.

[revised manuscript text omitted]

---

## Referee Comment (RC2) · John Idowu (Referee) · 28 Jun 2016

The usefulness of the study is in part dependent on the database of literature from which studies have been drawn. Limiting the literature database to the seven selected journals may have affected the outcome of the results. However, this limitation was acknowledged by the authors. Depending on the region, many specific soil health related studies are published in regional journals. It may be difficult to capture the efficacy of different soil health practices even in temperate regions without drawing on the regional/local journals that deal with specific soil health management issues. Nevertheless, the methodology of the authors is a good first step in the direction of identifying intervention practices that can improve soil health.

---

## Author Comment (AC3) · 4 Jul 2016

Dear John,

Thank you for your comment and for highlighting the importance of region-specific studies for our analysis, which we did not include. We appreciate that this is a limitation of our study, as you highlight, but we inevitably had to limit the number of journals we included in our analysis, given that we wanted to ensure that we carried out a comprehensive review of all papers, which takes a considerable amount of time. However, these journals included a wide range of soil related studies done in different regions and from different perspectives; as such, we believe that they are a representative sample of research done on enhancing and maintaining soil health. On top of this, we also used specific searches in Web of Science database, which identified a final 542

papers of relevance to soil health from a much wider range of journals. Given this, and the broad range of soil related studies from different regions of the world that our search identified, we feel that we managed to capture a representative range of relevant literature. Also, as we indicate in our discussion (line 6, page 10), there is scope to include a wider range of literature in the future, including from non-peer reviewed journals as you suggest.
* * *

---

## Author Comment (AC4) · 1 Aug 2016

Dear Saskia,

Thank you for your comments, especially as they highlight that we had not specified when the review was carried out. Here I respond to the larger issue of literature inclusion first, and then will respond to the other comments in order. The changes are marked in the attached manuscript.

While I appreciate that it would be good to have the most up-to-date literature for the review, the project and review were carried out in 2012, over a short timeframe, and the Delphi Assessment was carried out in 2013, hence the journals only being scanned up to 2012. I agree with your comment that the journal Land degradation and development is within the scope of this paper. However, at the time, we consulted with soil research

experts as to which journals to prioritise (page four line 26). As stated on page 12 line 36, we received suggestions for literature from the Expert Panel that was not captured by the review. The suggestions of the expert panel were not included as the review had been completed, and the amount suggested would have warranted another review. We outline this shortcoming on page 13 line four for the limitations of the paper, but also point out that despite this, a lot can be achieved over a short timeframe. Dates for when the review and assessment were carried out are now included in the methods section (page five line 15).

In answer to your request for additional results in the abstract, the beneficial practices found from the review are already mentioned in the abstract, on page one, line 36. I don't feel that it would be useful to repeat them within the abstract. On page two line 29, I have replaced the phrase "are being eroded" with "are deteriorating" as requested. I am more than happy to link to other papers being submitted for the special issue, and have cited the Keesstra et al paper in the manuscript where suggested (page two line 34).

Your fifth comment (page three) where you request more information on the benefits of better soil management, is largely answered in the previous paragraph where we state some of the problems occurring in the face of deteriorating soil health, and additions would likely result in repetition in this section. Thank you for the literature suggestions, but here we are trying to highlight that there are many unknowns to current practices on soil health and fertility.

In response to your comment on page five, we consulted farmers and land managers, and followed their recommendations. I have added a sentence to this effect (line 13). As explicitly mentioned in the Table 1 title on page six line 33, the practices have been grouped into indicative categories. These categories are then shown in the 'Category' column in the table itself. "Number of papers" has been added to the 'Y' axis in Figure 1.

In response to your comment on page 11 of the manuscript, if I have understood you correctly, finding out the number of places where soil management practices based on non-scientific knowledge are occurring, is beyond the scope of this paper. On page 12, line 21, we clearly state that our findings suggest that we do not yet have a full understanding of the consequences of practices on soil health. From line 21 onwards, we discuss the trade-offs of the practices, and on line 37 of the same page, we suggest that gleaning knowledge from practitioners is key to enhancing soil health. I have added a sentence (page 13 line three) to say "In addition, their knowledge would perhaps provide a wider range of practices for researchers to explore, widening the tools available to enhance soil health". In response to your final comment, we felt that it was more appropriate to address each limitation in turn, to show that we recognise them, but then to show where the limitations could lead to future opportunities in this area.

Overall I feel that we have clearly explained our methods and why we used them, and also that we were very upfront with the limitations of this paper and how the method can be improved in the future. The aim of this paper was not to review all the literature available on this subject, but to investigate whether the Delphi method could be used as a tool to help make decisions about practices affecting soil health, and we hope that the review will be used as a starting point for future reviews and assessments in this area.

Best wishes,

Georgina

Please also note the supplement to this comment:
http://www.soil-discuss.net/soil-2016-17/soil-2016-17-AC4-supplement.pdf

**Supplement:**

[revised manuscript text omitted]

---

## Author Response (AR1)

Dear Saskia,

Thank you for your comments.

In response to your first comment, the results have been moved to the latter part of the abstract
as requested (page 2, line 11). As previously explained, and also explained in the legend for
Table 1 (page 7, line 7), the practices are grouped into indicative categories, based on
categories of effectiveness by Sutherland et al. (2015): 1. Beneficial; 2. Likely to be beneficial;
3. Trade-offs between benefits and adverse side-effects; 4. Unknown effectiveness; 5.
Unlikely to be beneficial; 6. Likely to be ineffective or to have adverse side-effects.

Please also find an additional paragraph (page 11, line 11) giving more detail about practices
falling into the 'trade-offs' and 'unknown effectiveness.'

Best wishes,
Georgina

[revised manuscript text omitted]